# Forward Modeling for Partial Observation Strategy Games - A StarCraft Defogger

**Gabriel Synnaeve**[*]
Facebook, NYC
gab@fb.com

**Zeming Lin**[*]
Facebook, NYC
zlin@fb.com

**Jonas Gehring**
Facebook, Paris
jgehring@fb.com

**Dan Gant**
Facebook, NYC
danielgant@fb.com

**Vegard Mella**
Facebook, Paris
vegardmella@fb.com

**Vasil Khalidov**
Facebook, Paris
vkhalidov@fb.com

**Nicolas Carion**
Facebook, Paris
alcinos@fb.com

**Nicolas Usunier**
Facebook, Paris
usunier@fb.com

## Abstract

We formulate the problem of *defogging* as state estimation and future state prediction from previous, partial observations in the context of real-time strategy games. We propose to employ encoder-decoder neural networks for this task, and introduce proxy tasks and baselines for evaluation to assess their ability of capturing basic game rules and high-level dynamics. By combining convolutional neural networks and recurrent networks, we exploit spatial and sequential correlations and train well-performing models on a large dataset of human games of StarCraft[®]: Brood War[®][†]. Finally, we demonstrate the relevance of our models to downstream tasks by applying them for enemy unit prediction in a state-of-the-art, rule-based StarCraft bot. We observe improvements in win rates against several strong community bots.

## 1 Introduction

A current challenge in AI is to design policies to act in complex and partially observable environments. Many real-world scenarios involve a large number of agents that interact in different ways, and only a few of these interactions are observable at a given point in time. Yet, long-term planning is possible because high-level behavioral patterns emerge from the agents acting to achieve one of a limited set of long-term goals, under the constraints of the dynamics of the environment. In contexts where observational data is cheap but exploratory interaction costly, a fundamental question is whether we can learn reasonable priors – of these purposeful behaviors and the environment's dynamics – from observations of the natural flow of the interactions alone.

We address this question by considering the problems of state estimation and future state prediction in partially observable real-time strategy (RTS) games, taking StarCraft: Brood War as a running example. RTS games are multi-player games in which each player must gather resources, build an economy and recruit an army to eventually win against the opponent. Each player controls their units individually, and has access to a bird's-eye view of the environment where only the vicinity of the player's units is revealed.

Though still artificial environments, RTS games offer many of the properties of real-world scenarios at scales that are extremely challenging for the current methods. A typical state in StarCraft can be represented by a $512 \times 512$ 2D map of "walk tiles", which contains static terrain and buildings, as

---

[*]These authors contributed equally

[†]StarCraft is a trademark or registered trademark of Blizzard Entertainment, Inc., in the U.S. and/or other countries. Nothing in this paper should be construed as approval, endorsement, or sponsorship by Blizzard Entertainment, Inc.

well as up to 200 units per player. These units can move or attack anywhere in the map, and players control them individually; there are about 45 unit types, each of which has specific features that define the consequences of actions. Top-level human players perform about 350 actions per minute [1]. The high number of units interacting together makes the low-level dynamics of the game seemingly chaotic and hard to predict. However, when people play purposefully, at lower resolution in space and time, the flow of the game is intuitive for humans. We formulate the tasks of state estimation and future state prediction as predicting unobserved and future values of relevant high-level features of the game state, using a dataset of real game replays and assuming full information at training time. These high-level features are created from the raw game state by aggregating low-level information at different resolutions in space and time. In that context, state estimation and future state prediction are closely related problems, because hidden parts of the state can only be predicted by estimating how their content might have changed since the last time they were observed. Thus both state estimation and future state prediction require learning the natural flow of games.

We present encoder-decoder architectures with recurrent units in the latent space, evaluate these architectures against reasonable rule-based baselines, and show they perform significantly better than the baselines and are able to perform non-trivial prediction of tactical movements. In order to assess the relevance of the predictions on downstream tasks, we inform the strategic and tactical planning modules of a state-of-the-art full game bot with our models, resulting in significant increases in terms of win rate against multiple strong open-source bots. We release the code necessary for the reproduction of the project at https://github.com/facebookresearch/starcraft_defogger.

## 2   Related Work

Employing unsupervised learning to predict future data from a corpus of historical observations is sometimes referred to as "predictive learning". In this context, one of the most fundamental applications for which deep learning has proven to work well is language modeling, i.e. predicting the next word given a sequence of previous words [2, 3, 4]. This has inspired early work on predicting future frames in videos, originally on raw pixels [5, 6] and recently also in the space of semantic segmentations [7]. Similar approaches have been used to explicitly learn environment dynamics solely based on observations: in [8], models are being trained to predict future images in generated videos of falling block towers with the aim of discovering fundamental laws of gravity and motion.

Combining the most successful neural network paradigms for image data (CNNs) and sequence modeling (LSTMs) to exploit spatio-temporal relations has led to fruitful applications of the above ideas to a variety of tasks. [9] propose a LSTM with convolutional instead of linear interactions and demonstrate good performance on predicting percipitation based on radar echo images; [10] use a similar approach to forecast passenger demand for on-demand ride services; [11] estimate future traffic in Bejing based on image representations of road networks.

StarCraft: Brood War has long been a popular test bed and challenging benchmark for AI algorithms [12]. The StarCraft domain features partial observability; hence, a large part of the current game state is unknown and has to be estimated. This has been previously attempted with a focus on low-level dynamics, e.g. by modeling the movement of individual enemy units with particle filters [13]. [14] try to anticipate timing, army composition and location of upcoming opponent attacks with a bayesian model that explicitly deals with uncertainty due to the fog of war. [15] do not deal with issues caused by partial information but demonstrate usage of a combat model (which is conditioned on both state and action) learned from replay data that can be used in Monte Carlo Tree Search.

In the above works on StarCraft, features for machine learning techniques were hand-crafted with extensive domain knowledge. Deep learning methods, aiming to learn directly from raw input data, have only very recently been applied to this domain in the context of reinforcement learning in limited scenarios, e.g. [16, 17]. To our best knowledge, the only deep learning approach utilizing replays of human games to improve actual full-game play is [18], which use a feed-forward model to predict the next unit that the player should produce. They apply their model as a production manager in a StarCraft bot and achieve a win rate of 68% against the game's built-in AI.

## 3 Task Description

Formally, we consider state estimation and future state prediction in StarCraft to be the problem of inferring the full game state, $y_{t+s}$ for $s \geq 0$, given a sequence of past and current partial observations, $o_0, \ldots, o_t$. In this work, we restrict full state to be the locations and types of all units on the game map, ignoring attributes such as health. A partial observation $o_t$ contains all of a player's units as well as enemy or neutral units (e.g. resource units) in their vicinity, subject to sight range per unit type. Note that during normal play, we do not have access to $y_t$. We can however utilize past games for training as replaying them in StarCraft provides both $o_t$ and $y_t$.

We note that humans generally make forward predictions on a high level at a long time-scale – humans will not make fine predictions but instead predict a general composition of units over the whole game map. Thus, we propose to model a low-resolution game state by considering accumulated counts of units by type, downsampled onto a coarse spatial grid. We ignore dynamic unit attributes like health, energy or weapon cool-down. By downsampling spatially, we are unable to account for minute changes in e.g. unit movement but can capture high-level game dynamics more easily. This enables our models to predict relatively far into the future. Good players are able to estimate these dynamics very accurately which enables them to anticipate specific army movements or economic development of their opponents, allowing them to respond by changing their strategy.

StarCraft: Brood War is played on rectangular maps with sizes of up to $8192 \times 8192$ pixels. For most practical purposes it is sufficient to consider "walk tiles" instead which consist of $8 \times 8$ pixels each. In our setup, we accumulate units over $r \times r$ walk tiles, with a stride of $g \times g$; Figure 4 shows a grid for $r = 32$ and $g = 32$ on top of a screenshot of StarCraft.

For a map of size $H \times W$ walk tiles, the observation $o_t$ and output $y_t$ are thus a $H_{r,g} \times W_{r,g} \times C_u$ tensor, with $H_{r,g} = \lceil \frac{H-r}{g} \rceil$, $W_{r,g} = \lceil \frac{W-r}{g} \rceil$, and number of channels $C_u$ corresponding to the number of possible unit types. We use disjoint channels for allied and enemy units. Each element in $o_t$ and $y_t$ thus represents the absolute number of units of a specific type and player at a specific grid location, where $o_t$ only contains the part of the state observed by the current player. Additional static information $\tau$ includes (a) terrain features, a $H \times W \times C_T$ tensor that includes elements such as walkability, buildability and ground height, and (b) the faction, $f_{\text{me}}$ and $f_{\text{op}}$, that each player picks. Thus, each input $x_t = (o_t, \tau, f_{\text{me}}, f_{\text{op}})$

Additionally, we pick a temporal resolution of at least $s = 5$ seconds between consecutive states, again aiming to model high-level dynamics rather than small but often irrelevant changes. At the beginning of a game, players often follow a fixed opening and do not encounter enemy units. We therefore do not consider estimating states in the first 3 minutes of the game. To achieve data uniformity and computational efficiency, we also ignore states beyond 11 minutes. In online and professional settings, StarCraft is usually played at 24 frames per second with most games lasting between 10 and 20 minutes [19].

## 4 Encoder-Decoder Models

The broad class of architectures we consider is composed of convolutional models that can be segmented in two parts: an encoder and a decoder, depicted in Figure 1.

In all models, we preprocess static information with small networks. To process the static terrain information $\tau$, a convolutional network $E_M$ of kernel size $r$ and stride $g$ is used to downsample the $H \times W \times C_T$ tensor into an $H_{r,g} \times W_{r,g} \times F_T$ embedding. The faction of both players is represented by a learned embedding of size $F_F$. Finally, this is replicated temporally and concatenated with the input to generate a $T \times H_{r,g} \times W_{r,g} \times (F_T + F_F + C_u)$ tensor as input to the encoder.

The encoder then embeds it into a $F_E$ sized embedding and passes it into a recurrent network with LSTM cells. The recurrent cells allow our models to capture information from previous frames, which is necessary in a partially observable environment such as StarCraft, because events that we observed minutes ago are relevant to predict the hidden parts of the current state. Then, the input to the decoder $D$ takes the $F_E$ sized embedding, replicate along the spatial dimension of $o_t$ and concatenates it along the feature dimension of $o_t$.

The decoder then uses $D$ to produce an embedding with the same spatial dimensions as $y_t$. This embedding is used to produce two types of predictions. The first one, $P_c$ in Figure 1, is a global head

that takes as input a spatial sum-pooling and is a linear layer with sigmoid outputs for each unit type that predict the existence or absence of at least one unit of the corresponding type. This corresponds to `g_op_b` described in section 5.1.1. The second type of prediction $P_r$ predicts the number of units of each type across the spatial grid at the same resolution as $o_t$. This corresponds to the other tasks described in section 5.1.1. The $P_c$ heads are trained with binary cross entropy loss, while the $P_r$ heads are trained with a Huber loss.

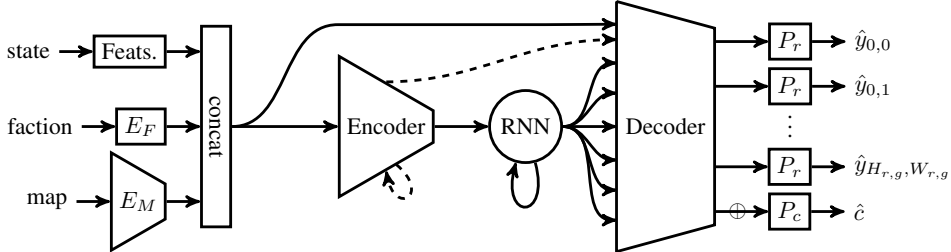

Figure 1: Simplified architecture of the model. Rectangles denote $1 \times 1$ convolutions or MLPs, trapezes denote convolutional neural networks, circles and loops denote recurrent neural networks. $\oplus$ denotes spatial pooling. The dashed arrows represent the additional connections in the architecture with the convolutional LSTM encoder.

We describe the two types of encoders we examine, a **ConvNet** (**C**) encoder and a **Convolutional-LSTMs** (**CL**) encoder.

Since the maps all have a maximum size, we introduce a simple **ConvNet** encoder with enough downsampling to always obtain a $1 \times 1 \times h$ encoding at the end. In our experiments we have to handle input sizes of up to $16 \times 16$. Thus, we obtain a $1 \times 1$-sized output by applying four convolutional layers with a stride of two each.

Our **CL** encoder architecture is based on what we call a spatially-replicated LSTM: Given that, for a sequence of length $T$, a convolution outputs a $T \times H \times W \times C$ sized tensor $X$, a *spatially replicated LSTM* takes as input each of the $X(:, i, j, :)$ cells, and encodes its output to the same spatial location. That is, the *weights* of the LSTM at each spatial location is shared, but the hidden states are not. Thus, at any given layer in the network, we will have $HW$ LSTMs with shared weights but unshared hidden states.

The **CL** encoder is made out of a few blocks, where each block is a convolution network, a downsampling layer, and a spatially-replicated LSTM. This encoder is similar to the model proposed in [20], with a kernel size of 1 and additional downsampling and upsampling layers. The last output is followed by a global sum-pooling to generate the $1 \times 1 \times F_E$ sized embedding required.

We also introduce skip connections from the encoder to the decoder in the **CL** model, where each intermediate blocks' outputs are upsampled and concatenated with $o_t$ as well, so the decoder can take advantage of the localized memory provided. With $k$ blocks, we use a stride of 2 in each block, so the output of the $k$-th block must be upsampled by a factor of $2^k$. Thus, only in the **CL** model do we also concatenate the the intermediate LSTM cell outputs to the input to the decoder $D$. These skip-connections from the intermediate blocks are a way to propagate spatio-temporal memory to the decoder at different spatial scales, taking advantage of the specific structure of this problem.

We use the same number of convolution layers in both **C** and **CL**.

# 5   Experiments

We hypothesize that our models will be able to make predictions of the global build tree and local unit numbers better than strong existing baselines. We evaluate our models on a human games dataset, and compare them to heuristic baselines that are currently employed by competitive rule-based bots. We also test whether we are able to use the defogger directly in a state-of-the-art rule-based StarCraft bot, and we evaluate the impact of the best models within full games.

## 5.1 Experiments On Human Replays

We define four tasks as proxies for measuring the usefulness of forward modeling in RTS games, and use those to assess the performance of our models. We then explain baselines and describe the hyper-parameters. Our models are trained and evaluated on the STARDATA corpus, which consists of 65,000 high quality human games of StarCraft: Brood War [19]. We use the train, valid, and test splits given by the authors, which comprise 59060, 3289 and 3297 games, respectively. Our models are implemented in PyTorch [21], and data preprocessing is done with TorchCraft [22].

### 5.1.1 Evaluation Proxy Tasks

In full games, the prediction of opponents' strategy and tactical choices may improve your chances at victory. We define tactics as *where* to send your units, and strategy as *what* kind of units to produce.

In strategy prediction, presence or absence of certain buildings is key to determining what types of units the opponent will produce, and which units the player needs to produce to counter the opponent. Thus, we can measure the prediction accuracy of all opponent buildings in a future frame.

We use two proxy tasks to measure the strength of defogging and forward modeling in tactics. One task is the model correctly predicting the existence or absence of all enemy units on the game map, correlated to being able to model the game dynamics. Another is the prediction of only units hidden by the fog of war, correlated to whether our models can accurately make predictions under partially observable states. Finally, the task of correctly predicting the location and number of enemy units is measured most accurately by the regression `Huber` loss, which we directly train for.

This results in four proxy tasks that correlate with how well the forward model can be used:

- `g_op_b` (global opponent buildings) Existence of each opponent building type on any tile.
- `hid_u` (hidden units) Existence of units that we do not see at time $t + s$, at each spatial location $(i, j)$, necessarily belonging to your opponent.
- `op_u` (opponent units) Existence of all opponent units at each spatial location $(i, j)$.
- `Huber` loss between the real and predicted unit counts in each tile, averaged over every dimension (temporal, spatial and unit types).

For the first three tasks, we track the F1 score, and for the last the `Huber` loss. The scores for `g_op_b` are averaged over $(T, C_u)$, other tasks (`hid_u`, `op_u`, `Huber` loss) are measured by averaging over all $(T, H_{r,g}, W_{r,g}, C_u)$; and then averaged over all games. When predicting existence/absence (`g_op_b`, `hid_u`, `op_u`) from a head (be it a regression or classification head), we use a threshold that we cross-validate per model, per head, on the F1 score.

### 5.1.2 Baselines

To validate the strength of our models, we compare their performance to relevant baselines that are similar to what rule-based bots use traditionally in StarCraft: Brood War competitions. Preliminary experiments with a kNN baseline showed that the rules we will present now worked better. These baselines rely exclusively on what was previously seen and game rules, to infer hidden units not in the current observation $o_t$.

We rely on four different baselines to measure success:

- *Previous Seen (PS)*: takes the last seen position for each currently hidden unit, which is what most rule based bots do in real games. When a location is revealed and no units are at the spot, the count is again reset to 0.
- *Perfect memory (PM)*: remembers everything, and units are never removed, maximizing recall. That is, with any $t_1 < t_2$, if a unit appears in $o_{t_1}$, then it is predicted to appear in $o_{t_2}$.
- *Perfect memory + rules (PM+R)*: designed to maximize `g_op_b`, by using perfect memory and game rules to infer the existence of unit types that are prerequisite for unit types that have ever been seen.
- *Input*: predicts by copying the input frame, here as a sanity check.

In order to beat these baselines, our models have to learn to correlate occurrences of units and buildings, and remember what was seen before. We hope our models will also be able to model high-level game dynamics and make long term predictions to generate even better forward predictions.

### 5.1.3 Hyperparameters

We train and compare multiple models by varying the encoder type as well as spatial and temporal resolutions. For each combination, we perform a grid search over multiple hyper-parameters and pick the best model according to our metrics on the proxy tasks as measured on the validation set. We explored the following properties: kernel width of convolutions and striding (3,5); model depth; non-linearities (ReLU, GLU); residual connections; skip connections in the encoder LSTM; optimizers (Adam, SGD); learning rates.

During hyperparameter tuning, we found Adam to be more stable than SGD over a large range of hyperparameters. We found that models with convolutional LSTMs encoders worked more robustly over a larger range of hyperparameters. Varying model sizes did not amount to significant gains, so we picked the smaller sizes for computational efficiency. Please check the appendix for a more detailed description of hyperparameters searched over.

### 5.1.4 Results

We report baselines and models scores according to the metrics described above, on $64$ and $32$ walktiles effective grids ($g$) due to striding, with predictions at 0, 5, 15, and 30 seconds in the future ($s$), in Table 1.

To obtain the existence thresholds from a regression output, we sweep the validation set for threshold values on a logarithmic scale from 0.001 to 1.5. A unit is assumed to be present in a cell, if the corresponding model output is greater than the existence threshold. Lower threshold values performed better, indicating that our model is sure of grid locations with zero units. Similarly, we fine-tune the existence threshold for the opponent's buildings. The value that maximizes the F1 score is slightly above 0.5. We report the results on the test set with the best thresholds on the validation set.

We note that for `g_op_b` prediction, the baselines already do very well, it is hard to beat the best baseline, PM+R. Most of our models have higher recall than the baseline, indicating that they predict many more unexpected buildings, at the expense of mispredicting existing buildings. On all tasks, our models do as well or better than baseline.

Our models make the most gains above baseline on unit prediction (columns `op_u` and `hid_u`). Since units often move very erratically due to the dynamics of pathfinding and rapid decision making, this is difficult for a baseline that only uses the previous frame. In order to predict units well, the model must have a good understanding of the dynamics of the game as well as the possible strategies taken by players in the game. For our baselines, the more coarse the grid size ($g = 64$, first row), the easier it is to predict unit movement, since small movements won't change the featurization. The results confirm that our models are able to predict the movement of enemy units, which none of the baselines are able to do. Our models consistently outperform both tasks by a significant amount.

In Table 1 the `Huber` loss gives a good approximation to how useful it will be when a state-of-the-art bot takes advantage of its predictions. During such control, we wish to minimize the number of mispredictions of opponent units. We average this loss across the spatial and temporal dimensions, and then across games, so the number is not easily interpretable. These losses are only comparable in the same $(g, s)$ scenario, and we do much better than baseline on all three accounts.

To give an intuition of prediction performance of our models, we visualized predicted unit types, locations and counts against the actual ones for hidden enemy units in Figure 2. We can see how well the model learns the game dynamics – in (a), the model gets almost nothing as input at the current timestep, yet still manages to predict a good distribution over the enemy units from what is seen in the previous frames. (b) shows that on longer horizons the prediction is less precise, but still contains quite some valuable information to plan tactical manoeuvres.

### 5.1.5 Evaluation in a Full-Game Bot

After observing good performance on metrics representing potential downstream tasks, we test these trained models in a StarCraft: Brood War full-game setting. We run a forward model alongside our

| Task: | op_u F1 | | | hid_u F1 | | | g_op_b F1 | | | Huber $\cdot 10^{-4}$ | | |
|---|---|---|---|---|---|---|---|---|---|---|---|---|
| $g : s$ | B | C | CL | B | C | CL | B | C | CL | B | C | CL |
| $64 : 15$ | 0.53 | 0.53 | **0.62** | 0.47 | 0.51 | **0.56** | 0.88 | 0.89 | **0.92** | 28.97 | 14.94 | **10.40** |
| $32 : 30$ | 0.33 | **0.48** | 0.47 | 0.26 | **0.44** | **0.44** | 0.88 | **0.92** | 0.90 | 1.173 | **0.503** | **0.503** |
| $32 : 15$ | 0.34 | 0.48 | **0.51** | 0.26 | 0.45 | **0.48** | 0.88 | 0.91 | **0.94** | 1.134 | 0.488 | **0.430** |
| $32 : 5$ | 0.35 | 0.44 | **0.52** | 0.27 | 0.38 | **0.47** | 0.89 | 0.90 | **0.95** | 1.079 | 0.431 | **0.424** |
| $32 : 0$ | 0.35 | 0.44 | **0.50** | 0.27 | 0.38 | **0.45** | 0.89 | **0.90** | 0.89 | 1.079 | **0.429** | 0.465 |

Table 1: Scores of our proposed models (C for ConvNet, CL for Convolutional LSTMs) and of the best baseline (B) for each task, in F1. The `Huber` loss is only comparable across the same stride $g$.

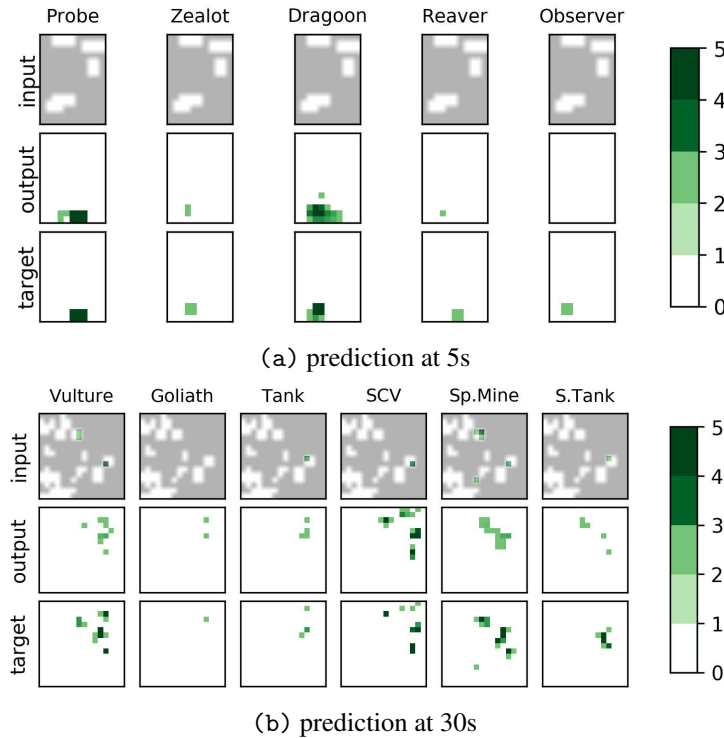

(a) prediction at 5s

(b) prediction at 30s

Figure 2: Enemy unit counts of the specified type per map cell, where darker dots correspond to higher counts. The top row of each plot shows the model input, where green indicates an input to our model. Grey areas designate the areas hidden by fog of war but are not input to our model. Middle row shows predicted unit distributions after 5 and 30 seconds. Bottom row shows real unit distributions.

| Win rate | | |
| --- | --- | --- |
| Normal Vision | 61 (baseline) | |
| | **Enhanced vision used for** | |
| | Tactics | Build | Both |
| Full Vision | 57 | 66 | **72** |
| Defog $t+0s$ | 57 | 62 | 59 |
| Defog $t+5s$ | 61 | **66** | 61 |
| Defog $t+30s$ | 50 | 59 | 49 |

Table 2: Average win rates with strategies that integrate predictions from defogger models but are otherwise unmodified.

| Win rate | | |
| --- | --- | --- |
| Normal Vision | 59 (baseline) | |
| | **Enhanced vision used for** | |
| | Tactics | Build | Both |
| Full Vision | 61 | 64 | **70** |
| Defog $t+0s$ | 61 | **63** | 55 |
| Defog $t+5s$ | 61 | **63** | 55 |
| Defog $t+30s$ | 52 | 51 | 43 |

Table 3: Average win rates from a subset of the games in the previous table. These games feature just one of our bot's strategies (focused on building up economy first) against 4 Terran opponents.

modular, rule-based, state-of-the-art StarCraft bot. We apply minimal changes, allowing the existing rules – which were tuned to win without any vision enhancements – to make use of the predictions.

We played multiple games with our bot against a battery of competitive opponents (see Table 5 in Appendix). We then compare results across five different sources of vision:

- *Normal*: Default vision settings
- *Full Vision*: Complete knowledge of the game state.
- *Defog + 0s*: Defogging the current game state.
- *Defog + 5s*: Defogging 5 seconds into the future.
- *Defog + 30s*: Defogging 30 seconds into the future.

We take the best defogger models in $op_u$ from the validation set. For each of the *Full vision* and *Defog* settings, we run three trials. In each trial we allow one or two of the bot's modules to consider the enhanced information:

- *Tactics*: Positioning armies and deciding when to fight.
- *Build Actions*: Choosing which units and structures to build.
- *Both*: Both Tactics and Build Actions.

In Tables 2 and 3, we investigate the effects of enhanced vision on our StarCraft bot's gameplay. For these experiments, our changes were minimal: we did not change the existing bot's rules, and our bot uses its existing strategies, which were previously tuned for win rates with normal vision, which essentially uses the *Previous Seen* baseline. Our only changes consisted of substituting the predicted unit counts (for *Build Actions*) and to put the predicted unseen units at the center of the hidden tiles where they are predicted (for *Tactics*). In our control experiment with *Full Vision*, we do the exactly the same, but instead of using counts predicted by defogging, we input the real counts, effectively cheating by given our bot full vision but snapping to the center of the hidden tiles to emulate the defogger setting. We run games with our bot against all opponents listed in Table 5, playing a total of 1820 games for each setting in Table 2.

Because our bot is Zerg, and most Zerg vs Zerg matchups depend much more on execution than opponent modeling, we do not try any Zerg bots. On average, over all match-ups and strategies, using the defogger model boosts the win rate of our rule-based bot to 66% from 61% (baseline), as Table 2 demonstrates. Overall, the defogger seems to hurt the existing Tactics module more than help it, but improves the performance of Build Actions.

We note that any variance in defogger output over time produces different kinds of errors in Build Actions and Tactics. Variance in inputs to Build Actions are likely to smooth out over time as the correct units are added in future time steps. Underestimations in Tactics cause the army to engage; subsequent overestimations cause it to retreat, leading to unproductive losses through indecision.

We broke down those results in a single match-up (Zerg vs. Terran), using a single strategy, in Table 3, the trends are the same, with encouraging use of the defogger predictions for Build Actions alone,

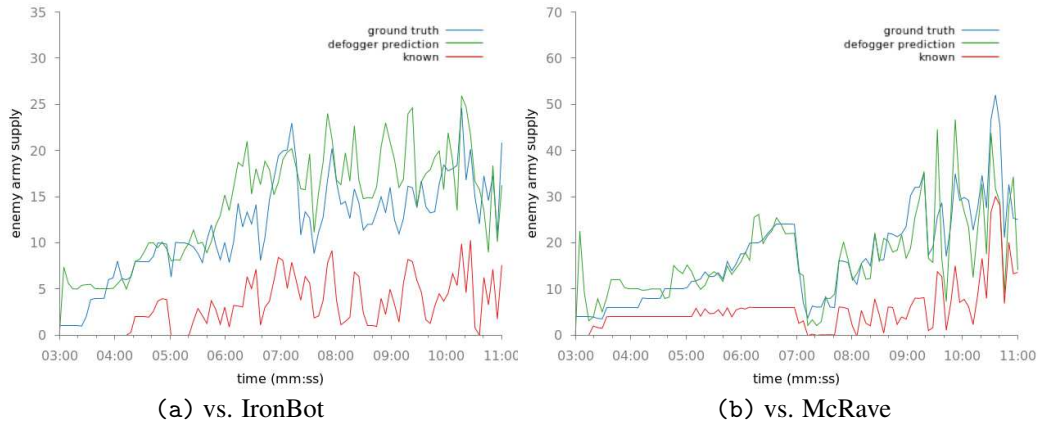

|                     |                    |
|:-------------------:|:------------------:|
| (a) vs. IronBot     | (b) vs. McRave     |

Figure 3: Plot showing the enemy army "supply" ($\approx$ unit counts) during the game: green is the ground truth, blue is the prediction of the defogger, and red is the known count our bot would normally have access with *Normal Vision*, equivalent to the *PS* baseline.

and poor result when combining this with Tactics. It suggests we could get better results by using a different defogger model for Tactics, or tuning the rules for the enhanced information conditions. Finally, in Figure 3, we observe that the defogger model is able to much more precisely predict the number of units in the game compared to using heuristics, equivalent to the *PS* baseline.

## 6 Conclusion and Future Work

We proposed models for state estimation and future state prediction in real-time strategy games, with the goals of inferring hidden parts of the state and learning higher-level strategic patterns and basic rules of the game from human games. We demonstrated via off-line tests that encoder-decoder architectures with temporal memory perform better than rule-based baselines at predicting both the current state and the future state. Moreover, we provide analysis of the advantages and pitfalls of informing a the tactical and strategic modules of a rule-based bot with a forward model trained solely on human data.

Forward models such as the defogger lack a model of how the agent acts in the environment. We believe the promising results presented in this paper open the way towards learning models of the evolution of the game conditioned on the players' strategy, to perform model-based reinforcement learning or model predictive control.

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
