[Supplementary Material]

# Appendix

| Model | Measure: stride ($g$) / step ($s$) | op_u | | | hid_u | | | g_op_b | | | Huber $\cdot 10^{-4}$ |
|---|---|---|---|---|---|---|---|---|---|---|---|
| | | P | R | F1 | P | R | F1 | P | R | F1 | |
| Input | 64 / 15 | 0.74 | 0.27 | 0.40 | 0.00 | 0.00 | 0.00 | 0.97 | 0.25 | 0.39 | 32.05 |
| PS | | 0.66 | 0.41 | 0.50 | 0.50 | 0.30 | 0.38 | 0.97 | 0.48 | 0.64 | 28.97 |
| PM. | | 0.43 | 0.68 | 0.53 | 0.43 | 0.53 | 0.47 | 0.94 | 0.70 | 0.80 | 37.41 |
| PM+R | | 0.25 | 0.70 | 0.37 | 0.20 | 0.57 | 0.30 | 0.95 | 0.82 | 0.88 | - |
| conv-lstm | | 0.53 | 0.75 | **0.62** | 0.47 | 0.67 | **0.56** | 0.94 | 0.91 | **0.92** | **10.40** |
| conv | | 0.48 | 0.61 | 0.53 | 0.46 | 0.58 | 0.51 | 0.91 | 0.86 | 0.89 | 19.43 |
| input | 32 / 30 | 0.44 | 0.15 | 0.23 | 0.00 | 0.00 | 0.00 | 0.97 | 0.29 | 0.45 | 1.173 |
| PS | | 0.32 | 0.34 | 0.33 | 0.25 | 0.26 | 0.25 | 0.97 | 0.65 | 0.78 | 1.269 |
| PM. | | 0.24 | 0.42 | 0.31 | 0.21 | 0.32 | 0.26 | 0.94 | 0.69 | 0.80 | 2.031 |
| PM+R | | 0.10 | 0.45 | 0.17 | 0.07 | 0.36 | 0.12 | 0.95 | 0.81 | 0.88 | - |
| conv-lstm | | 0.43 | 0.51 | **0.47** | 0.43 | 0.44 | **0.44** | 0.91 | 0.90 | 0.90 | **0.503** |
| conv | | 0.44 | 0.52 | **0.48** | 0.44 | 0.45 | **0.44** | 0.94 | 0.90 | **0.92** | **0.503** |
| input | 32 / 15 | 0.50 | 0.15 | 0.22 | 0.00 | 0.00 | 0.00 | 0.97 | 0.25 | 0.39 | 1.134 |
| PS | | 0.32 | 0.36 | 0.34 | 0.24 | 0.29 | 0.26 | 0.97 | 0.64 | 0.77 | 1.254 |
| PM. | | 0.22 | 0.47 | 0.30 | 0.19 | 0.38 | 0.25 | 0.94 | 0.70 | 0.80 | 2.443 |
| PM+R | | 0.09 | 0.50 | 0.16 | 0.07 | 0.41 | 0.12 | 0.95 | 0.82 | 0.88 | - |
| conv-lstm | | 0.47 | 0.55 | **0.51** | 0.47 | **0.49** | 0.48 | 0.96 | 0.92 | **0.94** | **0.430** |
| conv | | 0.45 | 0.52 | 0.48 | 0.44 | 0.45 | 0.45 | 0.93 | 0.90 | 0.91 | 0.488 |
| input | 32 / 5 | 0.63 | 0.15 | 0.24 | 0.00 | 0.00 | 0.00 | 0.98 | 0.21 | 0.34 | 1.079 |
| PS | | 0.32 | 0.39 | 0.35 | 0.23 | 0.31 | 0.27 | 0.97 | 0.61 | 0.75 | 1.196 |
| PM. | | 0.18 | 0.53 | 0.27 | 0.16 | 0.45 | 0.24 | 0.94 | 0.71 | 0.81 | 3.285 |
| PM+R | | 0.08 | 0.57 | 0.15 | 0.07 | 0.48 | 0.12 | 0.95 | 0.83 | 0.89 | - |
| conv-lstm | | 0.49 | 0.56 | **0.52** | 0.46 | 0.49 | **0.47** | 0.96 | 0.93 | **0.95** | **0.424** |
| conv | | 0.49 | 0.55 | **0.52** | 0.46 | 0.48 | **0.47** | 0.94 | 0.89 | 0.91 | 0.431 |
| input | 32 / 0 | 0.63 | 0.15 | 0.24 | 0.00 | 0.00 | 0.00 | 0.98 | 0.21 | 0.34 | 1.079 |
| PS | | 0.32 | 0.39 | 0.35 | 0.23 | 0.31 | 0.27 | 0.97 | 0.61 | 0.75 | 1.196 |
| PM. | | 0.18 | 0.53 | 0.27 | 0.16 | 0.45 | 0.24 | 0.94 | 0.71 | 0.81 | 3.285 |
| PM+R | | 0.08 | 0.57 | 0.15 | 0.07 | 0.48 | 0.12 | 0.95 | 0.83 | 0.89 | - |
| conv-lstm | | 0.48 | 0.52 | 0.50 | 0.45 | 0.45 | 0.45 | 0.94 | 0.85 | 0.89 | 0.465 |
| conv | | 0.50 | 0.54 | **0.52** | 0.48 | 0.47 | **0.47** | 0.94 | 0.88 | **0.91** | **0.429** |

Table 4: Performance of our proposed models and all considered baselines for each task on F1, precision, and recall.

## Model Hyperparameters

Because there was little prior research on the types of models that would work well for this task, we arrived to our final models through a combination of hand tuning and random grid search. The hyperparameters we tuned over were:

- Model structure:
  - A purely convolutional encoder, described as **ConvNet** in Section 4.
  - A convolutional encoder with spatially replicated LSTMs, described as **Convolutional-LSTMs** in Section 4.
  - A encoder decoder model without an RNN, with convolutions in the temporal dimension as well. This did not work as well as models with an LSTM, in preliminary experiments.
- Model depth - 4, 9, or 16 layers. In the convolutional-LSTM models, this corresponds to 1, 2, or 3 extra LSTM layers.
- Whether we predict a delta from the previous frame or the full value.

- Whether there is a skip connection that bypasses the encoder or not.
- Huber vs MSE loss.
- Optimizer parameters, including learning rate [1e-2, 1e-3, 1e-4, 1e-5], SGD vs. Adam, decaying the learning rate, momentum [0, 0.9, 0.99].
- Basic ConvNet blocks being either [basic convolution, gated convolution, residual block].
- Nonlinearity [ELU, ReLU, SeLU].

The convolutions were fixed to have 128 channels at all layers, and the LSTM was fixed to have 256 channels whenever they occur. The **ConvNet** structure had 300k parameters at 4 layers and 600k at 9 layers. The **Convolutional-LSTM** had 450k at 4 layers and 800k at 9 layers. We noticed that the number of layers did not seem to impact results, so we believe that the number of parameters was not a big factor in comparing the different model structures. In the final experiments, we fixed predicting a delta over previous frame, huber loss, Adam with learning rate 1e-4, and an ELU nonlinearity. We swept over [4, 9] layers, and [basic convolution, gated convolution, residual block], and picked the best performing hyperparameters for each structure to report in the table.

**Complexity of StarCraft**

StarCraft is a Real-Time Strategy game, as such: it has approximately 24 turns/second and units can take simultaneous moves. We make some approximations to try to compute the complexity of the game. Each unit can perform thousands of actions per frame, i.e. move to each location on the map. If we consider actions to be the same if they create the same trace in the next few frames, we can estimate that there might be 25 different actions per unit over a short time span. We also approximate 50 units for a given player, producing a branching factor $b = 25^{50}$. The depth of the game corresponds to the number of frames, and the average length of high level of skill human games is approximately 15 minutes [19], $d = 21600$. Thus, we give a back-of-the-envelope possible number of games at $(25^{50})^{21600}$ [23, 24].

The size of the state space is equivalently gigantic, let us assume that $512 \times 512$ is an adequate resolution to play at the highest level of play. There are various elevations and "walkability" of terrain, as well as a limit on the number of units a player can have between 100 and 400. Those units have static (range, speed, attack, armor, etc) and dynamic (location, hit points, cooldown, energy, velocity, acceleration, etc) properties. For the sake of argument, we assume 15 continuous properties that we can bin into 100 bins each, and 50 discrete properties of 2 values each, being the 50 or so flags are that exposed in BWAPI. The state space is then around $10^{45}$ per unit. As we assumed 50 units on

Figure 4: Screenshot of a replay viewed in StarCraft: Brood War. The thin white lines are marking squares of 32 by 32 walk tiles for which we predict accumulated unit counts by type. During normal play, fog of war (not shown) would prevent the orange player on the bottom right from observing the highlighted building on the upper left hand corner due to limited vision of their units.

each side, this gives a back of the envelope calculation of $100^{10^{45}}$ states. Of course, many of these states are highly improbable and/or hard to reach.

Since we ignore the attributes and only focus on the locations, and we do somewhat aggressive binning, the problem that we present in this paper is far less complex. Each frame of the dataset has 230 channels, representing each unit type - 115 different units and 2 sides. We are to predict to non-partially observed forward prediction of each unit type in each spatial location. At a maximum map size of 512 and grid size of 32, Each frame is thus $16 \times 16 \times 230$, and the output is of the same size. Thus, each frame has 58880 regression problems – the number of units in each spatial cell. We also have approximately "60k games $\times$ 70 time steps $\times$ 2 sides" $= 8.4$ million frames. This dataset is on the same order of magnitude as ImageNet, with an output structure comparable to image segmentation.

| Name | StarCraft Race | Achievement |
|------|----------------|-------------|
| IronBot | Terran | 1st Place, AIIDE 2016 |
| LetaBot | Terran | 1st Place, SSCAIT 2017 |
| McRave | Protoss | 4th Place, SSCAIT 2018 |
| Skynet | Protoss | 1st Place, AIIDE 2011 + 2012 |
| tscmoo | Terran/Protoss | 1st Place, AIIDE 2015 |
| UAlbertaBot | Terran/Protoss | 1st Place, AIIDE 2013 |

Table 5: Set of bots used as opponents in our full-game experiments.