[Reviews · NeurIPS 2018]

Reviewer 1



This paper focuses on real-time strategy games, and presents a model to make predictions over the parts of the game state that are not observable, as well as predicting the evolution of the game state over time. The proposed model is based on a encoder/decoder architecture that integrates convolutional networks with recurrent neural networks. This is an interesting paper with promising results. The most interesting part for me is that the proposed model seems to me a "starting point", and this opens up a very interesting avenue of research for the future. For example, the current module provides a prediction, but could it be used to provide a distribution over the possible game states, from where we can sample? This would be very useful for forward planning, for example. Also, could we feed a plan, and use the de-fogger as a forward model, of the execution of the plan? etc. So, in summery, very interesting work, showing a lot of potential! Some specific comments: - "To our best knowledge, the only deep learning approach utilizing replays of human games to improve actual full-game play is [17]" -> in "Evaluating Real-Time Strategy Game States Using Convolutional Neural Networks" by Barriga et al. they use deep learning to learn an evaulation function to be used in a game tree search framework for full game play in RTS games (in microRTS, not StarCraft though). - "downsampled onto on a coarse spatial grid" -> "downsampled onto a coarse spatial grid" - Section 4: this section is a bit foncusing to me, and I think it could have been written in a more clear way. For example, when describing the "two types of encoders", are both used in the model? [edit: this is clarified later in the experiments, but it shuold be clear from the start] or are they two alternatives that were compared? - Section 5: - "The full games, the prediction of" -> "In full games, the prediction of"? - "Varying model sizes did not amount to significant gains, so we picked the smaller sizes for computational efficiency." -> what sizes were selected for final experiments? - Concerning the baselines, the three first ones are basically the same exact ones used by Uriarte et al. in "Single believe state generation for partially observable real-time strategy games", so, this makes me think that if the last paragraph of the future work section is successful, the de-fogger could be used not just in RL, but also as a forward model in tree search algorithms. - About the results (Table 1): I find it curious that with the ConvNet encoder, results seem to improve when increasing time (s)!! (Except for the Huber loss) Do you have an explanation for that? I would expect results to DEGREASE over time! - Also, about the use of Huber loss: wouldn't it be more interesting to see error over time rather than average over the time dimension? i.e. see how error grows as the amount of time we are the model to predict into the future grows? - Results show that incorporating the de-fogger into an existing StarCraft bot helps a bit, but it mostly hinders, since the existing modules are not expecting it. What kind of changes would be required into the bot to make it take full advantage of the defogger? - I would have liked to see how performance degrades/improves when spatial resolution is changed. The current experiments show for g = 32 and 64. But what happens when you ask the model to do finer grained predictions? e.g. g = 16, or even g = 8? Does it introduce a lot of noise?

Reviewer 2



Update after rebuttal: the new details added by the authors in the rebuttal helped clarify some of the doubts I have, so I have changed my score accordingly. -------------------------------- Summary: The paper presents a deep learning approach to predict some hidden features in a StarCraft game, such as existence of opponent building types and unit types at any location and time in the game. The paper suffers from lack of details on both the game, approach and the experiment setup, so it is quite hard to read. More Strengths: The paper presents new ideas in applying deep learning in the context of strategy games, which is interesting. Weaknesses: The paper does not provide enough details on the game and various concepts in the game to justify the design of the proposed infrastructure. Due to the lack of sufficient details on the game to justify and inform the design of the deep network architecture, it is hard to convince readers why the approach works or does not work. The reproducibility is therefore affected. The take-away from the paper or generalization beyond StarCraft is also limited. Other general comments: - Tables and figures should appear near where they are mentioned, so Table 4 and Figure 4 should not be in the appendix. - Too many papers referenced are from non-peer reviewed websites, such as Arxiv and CoRR, weakening the paper in terms of argumentation. The authors are highly encouraged to replace these papers by their peer-reviewed counterparts - The selected baselines are too simple. They are all simple, rule-based, and myopic, with no attempt to learn anything from the provided training data, which the authors’ architecture makes use of. It would be a fairer comparison should authors have tried to design a little more sophisticated baselines that incorporate some form of learning (via classical prediction models, for example). Other detailed comments: - Various statements about the game and how players behave/act in the game miss explanation and/or supporting reference * Page 1, line 31: “Top-level human players perform about 350 actions per minute“ * Page 3, line 86: “... humans generally make forward predictions on a high level at a long time-scale; ...” * Page 6, line 240-241: “Since units often move very erratically...”: not really, units often move according to some path. This also illustrates why the selected baselines are too weak; they make no use of the training data to try to infer the path of the moving units. - Page 2, line 81: Full state is not defined properly. What are included in a full state? - Page 3, line 96: what are “walk tiles”? - Page 3, line 119: What is “the faction of both players”? - (minor) Figure 4 does not print well in black and white - Page 3, line 131-132: The author refers to the g_op_b task that is introduced later in the paper. It would be better if the authors can describe such concepts in a section devoted to description of the games and related tasks of interest. - Page 4, line 139: “h” is not defined yet. - Page 5, line 208-210: What game rules were used for this baseline? In general, how do game rules help? - Page 7, line 271-276: This part needs more elaboration on what modules are available in the bots, and how they can use the enhanced information to their advantage in a game. - Page 7, line 286-287: The reason to exclude Zerg is not convincing nor well supported externally. The author is encouraged to include a Zerg bot regardless, then in the discussion section, explain the result and performance accordingly.

Reviewer 3



This is an acceptable paper. The development of a forward model to partially observable problems is a complex and important task and I think the research is valuable. The paper proposes an architecture which is well studied in NLP but relatively new for games. However, I'm about the motivation of the architecture of the network. It seems very complicated and is complicated to know the reason of several uses. Also, I think the baselines used are rather simple. Better baselines would be desirable. Finally, I don’t see the link to access to the code. Further comments: Introduction Line 37 “assuming full information at training time” Not sure about this statement. The use of full information to predict PO is a valid point for StarCraft forward models? Line 45- “better than baselines” The baselines are rather simple. Related work I like the general introduction about predictive learning, which clearly lays out the importance of the research area. Task Description Line 99-101 The formulation of the feature set is not very clear. Encoder-Decoder model This is arguably the most interesting part. Using this architecture in video game environments is novel. The architecture seems very complex and the training time long. It is not clear why the sub neural network is used. Granted, empirically it seems that increase the features used to predict (e.g map, state). However, I think it is mandatory to compare several architectures of encoder-decoder to see the value of the proposed DNN. Line 152-158 I think this is a good idea. Experiments The code should be made available. Evaluation Proxy task I have some doubts about lines 179-181. Is this enough for the evaluation? A statistical analysis is recommended. Hyperparameters The selection of the paraments is poor. Optimization of the parameters could be an interesting improvement of the paper. Also, I think the authors should explain better and increase the uncertainty (I mean number of paramenters) to “optimize”. Results Line 230-233. The regression model could give a high error for the threshold. The baselines are too simple.